# Education—Migration Nexus: Understanding Youth Migration in Southern Ethiopia

**Tesfaye Semela [1,2,*] and Logan Cochrane [1,3]** 

[1]    Institute of Policy and Development Research, Hawassa University, P.O. Box 05, Hawassa, Ethiopia;
       logan.cochrane@gmail.com
[2]    Social and Cultural Sciences- IfE, Justus-Liebig University of Giessen, 35394 Giessen, Germany
[3]    Global and International Studies, Carleton University, Ottawa, ON K1S 5B6, Canada
*      Correspondence: Tesfaye.Semela-Kukem@erziehung.uni-Giessen.de

**Abstract:** The purpose of this study is to unravel the education–migration nexus in the African context, specifically Ethiopia. It examines why young people terminate their education to migrate out of the country. The study applies de Haas' aspiration—capability framework and Turner's macro, meso and micro sociology as its analytical lenses. It offers unique insight into the terrain of youth migration in southern Ethiopia based on empirical data obtained from two rural sub-districts known for high levels of youth out-migration. Data are generated based on interviews with would-be migrant youth, parents, teachers and school principals. The findings reveal that education has both direct and indirect impacts on youth migration. On the other hand, the results indicate that though terminating school could have negative ramifications on human capital accumulation at micro and macro levels, migration can positively impact households and local communities through investments made by individual migrants, migrant-returnees, and remittance-receiving households in small businesses or community development projects, which included better resourced schools.

**Keywords:** capability; education; migration; Ethiopia

## 1. Introduction

Migration is contentious, and the responses to the movement of people vary widely, as is the information shared about it. On the one hand, the displacement of people is at a record high, nearly 70 million people [1], and international migration is increasing, an estimated total of 244 million people globally [2]. However, the vast majority of people around the world remain in their countries of birth, with the International Organization for Migration [2] finding that only 3.3% of the world's population are international migrants. The response to migration has varied greatly. Some governments are enabling new migration options while others are seeking to stop potential migrants at their places of origin, along their routes of travel and at points of entry. The 2030 Agenda and the Sustainable Development Goals frame migration as a contribution to sustainable development, and conceptualize migration as a cross-cutting issue, with specific references to migration made in the indicators and targets of 11 of the 17 goals [3].

This paper critically evaluates the link between education and migration. Specifically, it contributes to a growing set of literature analyzing this relationship, often highlighting the contextual factors involved [4–10]. Amidst the diversity of findings, however, there is a common trend: the question tends to be one of understanding the education–migration relationship of individuals from the Global South migrating to the Global North, and, more specifically, migrants from Africa and the Middle East. In this study, we contribute to the research landscape that seeks to understand how education shapes decisions to migrate and why young people discontinue their education for migration. Specifically, we

look at southern Ethiopia, where communities have experienced the loss of many young lives in their pursuit of relocating to other countries.

The present study employs [11] aspiration–capability framework to understand the education–migration nexus in southern Ethiopia. We examine the reasons behind young peoples' decisions to give up school at micro, meso, and macro levels. The findings inform decision makers regarding (re)designing strategies to reduce the negative ramifications of irregular migration. The depth of qualitative data also informs future research, particularly what measures ought to be considered when conducting quantitative studies.

## 2. Context: Education—Migration Nexus

Theories about migration have arisen from a range of disciplinary foundations, largely demography, economics, geography and sociology, which are informed by their respective philosophical traditions (e.g., functionalism, historical-structuralism, symbolic interactionism; see: [12]). Predominant amongst the theories, according to [12], are those rooted in functionalist thought. One example of this is the neo-classical economic perspective, viewing migration choices as a reflection of rational cost–benefit analyses, and thereby focus on factors such as wage differentials [4,13,14]. A second example conceptualizes migration as a collective decision, aimed at diversifying income (as opposed to individual maximization) [15,16]. Other theorists explore the role of social networks and push–pull interactions (for detailed review, see: [12]).

The historical-structural perspectives draw on neo-Marxist, conflict theory. This tradition construes the causes and consequences of migration as an outcome of structural inequality resulting from class differences. The widely employed paradigms within this tradition are world-systems theory and dependency theory [12]. The world-systems perspective is grounded in the concepts of structural penetration and 'imbalancing' of peripheral areas creating the conditions for mass displacements out of them [17,18]. It views migration as a corollary of economic globalization and market penetration across national boundaries [19]. The world systems theory addresses, at least partially, the first problem with the dependency model in explaining existing social relations in the source countries. Building on Wallerstein [19], this model posits that "the penetration of capitalist economic relations into peripheral, non-capitalist societies creates a mobile population that is prone to migrate abroad" [20] (p. 444). According to this theory, migration is a natural outgrowth of the dislocations and disruptions caused by capitalist development. With the expansion of capitalism over larger parts of the world, the influence and control of the market is also extended over land, raw materials and labor within the peripheral regions, creating a vast mobile population. The material and ideological links along with the investment capital, according to Sussen [21], usher in a massive influx of people from developing countries to major cities in developed countries to take up low-paying jobs.

Dependency theory argues that labor migration is stimulated by the uneven spatial development resulting from colonial and neocolonial political and economic relationships between the developed capitalist economies and their underdeveloped peripheries. As such, migration is viewed not only as a response to the existing imbalanced situation, but also a social process that reinforces it. Migration, as Amin [22] argues, represents a spatial transfer of value greater than the return to the individual in remitted wages because it selectively captures only the most productive and educated workers from the underdeveloped countries or regions. This view is best represented by the dual labor market theory [23] argues that international labor migration is caused by the voracious appetite for immigrant labor inherent in the economic structure of developed nations.

None of these theories provide sufficient understanding of migration within the African continent and the Middle East. In effect, the theory building effort evoked heavy criticism not only for adopting "a one-size fits all" approach, but also for oversimplifying the complexities involved in conceptualizing migration [12,24]. To address the apparent theoretical pitfalls, de Haas and Carling [12] called for an eclectic approach that allows for the use of a combination of existing migration theories despite their allegiance to contending paradigms. In the paragraphs that follow, we will briefly review the

aspiration–capabilities framework [11,12] that purports to provide more comprehensive conceptual tools not only to make sense of the antecedents and consequences of migration, but also establishing the theoretical link between migration and education.

Education is conceived to drive migration in different ways. For example, acquiring a higher educational qualification may result in new migration opportunities and aspiration, due to the perceived competitiveness for employment abroad [25]. There is a significant amount of literature that assesses the flows of people following this type of migration, as well as the questions arising from it [26–28]. The flows of migration of this type tend to be higher skill, but are not necessarily from the Global South to the Global North, as they may also include South–South migration patterns; however, the literature has focused more on the former (for alternatives, see: [29]. In these conceptualizations, education is a positive driver [30]. However, there are also drivers that relate to education in different ways. For example, education might not be accessible, of high quality, or be viewed as leading to viable alternatives, and thereby contribute to discontinuation. Individuals who discontinue education may seek formal or informal options of low skill employment in their locality, nationally or internationally. In the case of Ethiopia, while the majority of migrants relocate domestically, there are some sectors that have attracted large numbers of formal migrants to relocate internationally, such as low-skill workers going to the Middle East. In these instances, migration is not only driven by opportunity abroad, but also by vulnerability in their place of origin. The work opportunities that are obtained are often precarious, low-paid and operate with restrictive agreements. We contribute to the latter of these drivers, as we view less evidence is available regarding it.

*Education as Capability*

The existing literature highlights education in particular, as it constitutes the key aspects of the capability approach [18,25,31–33]. Specifically, theories that attempt to unpack the antecedents of international labor migration posit education as one of the important drivers [11]. In the paragraphs that follow, we briefly explore Amartya Sen's notion of capabilities vis-à-vis education, and how the latter is substantively, if not empirically linked to migration.

Sen [34] (p. 87) defines capabilities as 'the alternative combination of functionings that are feasible for [a person] to achieve'; they are 'the substantive freedom' a person has 'to lead the kind of life he or she has reason to value'. Capabilities are the various functionings that a person can attain—whereas functionings are the constitutive elements of living, that is, doing and being [33]. Examples of functionings, according to Robeyns [33] (p. 76), include " . . . being healthy, being educated, holding a job, being part of a nurturing family, having deep friendships, etc."

Over the past two decades, there has been increased scholarly interest aiming to understand the role of education within the capabilities approach (CA) [18,31,33,35]. These attempts include locating the place of education in defining the well-being or freedoms of individuals or collectives in terms of living the life they reason to value. In this regard, Nussbaum [36] identifies education as a core capability of people's capability set. Saito [18] and Robeyns [33] demonstrate the superiority of the CA over other alternative theories, especially, the human capital theory and the human rights approach to education. The human capital theory considers the *instrumental* aspect of education (education is viewed as means of human capital accumulation, in order to bring about economic benefits) without due consideration for its intrinsic aspects (i.e., education is not necessarily for economic benefits, but to achieve the freedom to live a life people reason to value).

Saito [18] further elaborates how education expands human wellbeing (capabilities) i.e., the freedom to achieve valued functionings [34]. He goes on to illustrate the role of education as extending capabilities as: "learning mathematics is an achievement in its own right; beyond that the knowledge s/he acquired, the learning mathematics gives the child the opportunity to become a mathematician, a physicist, or a banker" [18] (p. 27). In that sense, education constitutes both the instrumental and intrinsic values. This appears to fit into what Robeyns [33] characterized as the "instrumental personal economic role of education", which, she argues, "can help a person to find a job, to be less vulnerable

on the labour market, to be better informed as a consumer, to be more able to find information on economic opportunities, and so forth" (pp. 70–71). Nevertheless, warns Saito [18], the instrumental personal economic role of education may only come to fruition if " … the education provided is of minimally acceptable quality … " (p. 71).

Apart from inequalities that could result from the failure to provide a minimally acceptable quality of education for citizens, other sources of inequality could curtail opportunities because all individuals (for example, a child in remote rural area and his/her counterpart in urban center) may not enjoy equal opportunity. Nussbaum [37] in this regard contends: "Equality of resources falls short because it fails to take account of the fact that individuals need differing levels of resources if they are to come up to the same level of capability to function. They also have differing abilities to convert resources into actual functioning" (p. 35). As such, education (as a capability) is considered as the means to removing obstacles in the life of young persons "so that they have more freedom to live the kind of life which, upon reflection, they have reason to value" [38] (p. 3). According to de Haas [11,12], this includes the decision to migrate out of one's country of origin. Thus, within the capability framework, de Haas [12] (p. 24) establishes the substantive link between education and migration. He argues that education, in particular in rural areas, expands skills and knowledge, but also "people's awareness of alternative, consumerist, and urban lifestyles. This may change people's notions of the good life, and they may subsequently start to aspire to migrate."

From the forgoing discussion, it is evident that none of the theories of migration are able to provide comprehensive answers as to what underpins youth migration [11]. Analyzing the existing theories, one can decipher the presence of three major constellations; namely micro-level (e.g., individual choice), meso-level (social network or family theories), and macro-level (global) theories. However, none of these levels can single-handedly explain why young people migrate. As Turner [39,40] contends, as a sociological phenomenon, youth migration may be better understood if agency (micro) and structure (macro) are integrated as one accounts for the meso-level social reality. This is why, according to de Haas [11], the agency and structure approach adopted by various theoretical paradigms (i.e., functionalism, historical-structural and neo-Marxism) ignore the complex set of factors that come into play to explain the systemic relationship between migration and development.

## 3. Mobility of Ethiopians Internationally

The international movement of Ethiopians beyond their national borders is not a recent phenomenon, even though the scale of movements has risen significantly in recent decades. One of the notable contemporary movements of Ethiopians outside of their country followed the 1974 revolution and the protracted conflict that followed (domestically and internationally). Political dissidents and the intelligentsia sought opportunities and safety elsewhere, while ordinary people who opposed the system fled persecution and reprisal. As the military government (1974–1991) was toppled, new insecurities emerged (both physical and political), resulting in waves of movements in the late 1980s and early 1990s [40]. Although the historical record is largely absent, it appears that most international relocations from Ethiopia up until this point were due to internal push factors, as opposed to external pull factors. In more recent decades (mid-1990s onward), facilitated by policy changes, the improvement of transportation infrastructure and better access to information and communication technologies, Ethiopians have also relocated seeking work opportunities. The two types of destinations align with the conceptualizations mentioned above: (1) higher skilled migration to countries of the Global North, such as the United States, Italy, Canada and Sweden, which includes individuals seeking opportunities as well as individuals fleeing from persecution [40,41]; and (2) lower skilled migration to the Global South, specifically to the Middle East and South Africa [30,42,43]. The latter of these has experienced more irregular forms of movement than the former, although data is limited [29].

The challenges of irregular international movement have attracted broad attention, particularly regarding losses of lives. The sinking of boats on route to the Middle East [44], the murder of Ethiopians on route via land [45], and the suffocation of people in unsafe transportation conditions [46], are

examples of this. The routes are havens for criminals, with reports of physical and sexual violence as well as abduction and required ransom payments [47]. The Government of Ethiopia, along with its partners, has sought to inform the public of these dangers in an effort to reduce irregular movements. While the violence and risks cannot be compared, formal migration of low skilled workers is also highly precarious, with workers often hidden and threatened with deportation [48]. What drives the continued migration of people within this context is what this research set out to address. In focusing upon education, we recognize that decision-making is far more complex. This research is thus an attempt at offering insight into one component of a complex issue.

## 4. Materials and Methods

Within Ethiopia, there are regions where formal and irregular migration is more common than others. For example, migration to the Gulf Cooperation Countries is a common practice of youth in Tigray regional state while, in parts of the Southern Nations, Nationalities and Peoples' (SNNP) regional state there are higher rates of migration to South Africa (mostly through Kenya, Tanzania, Malawi, Mozambique) and the Middle East [49,50]. There are other locations from which migrants originate (Amhara and Oromia regional states), but limited data is available about the destinations [50]. This study focuses upon two specific locations within SNNP regional state, namely Kembata-Tembaro and Hadiya zones. Of interest to this study is analyzing the linkages between education and migration. The research questions we set out to answer were: (1) What is the role of education in influencing decision-making for youth to consider migrating out of the country? (2) Does education contribute to increasing aspirations and capability in migration decision-making?

We applied de Haas' [11] aspiration–capability framework to evaluate if education is enabling youth to achieve the functionings required: (a) to take advantage of opportunities available to them domestically; and, (b) the extent to which education, as a core capability [37,38], is impacting (positively or negatively) young peoples' migration aspirations. We closely examine these questions by specifically focusing on young people who aspire to migrate out of the country terminating their education. Since migration is a sociological phenomenon, we conceptualized the interaction between aspiration and capability in shaping migration plans (or actual decision to migrate for that matter) using Turner's [38–40] integrated sociological theory as our analytical lens. Using this integrated approach, we account for the micro-meso-macro levels of social realities. The selection of different categories of respondents (i.e., students, teachers, school principals, parents, and other knowledgeable community members) was meant to understand how the decision to migrate by an agent (individual migrant) is viewed and judged at micro, meso, and macro levels of social realities.

More specifically, the micro-meso-macro analysis enables us to explore who makes the decision to terminate education and migrate. This approach allows us to assess what roles the individual migrant, his/her parents and members of the extended family, neighbors, the village community, and the local governance practices play. It also enables a consideration of the national socio-economic and political (macro) context, which influences the migration aspiration and migration capabilities in decision-making. Drawing on Turner's [39,40] macro-meso-micro theories of sociology, we defined the levels of social realities as follows:

- Micro-level: The micro level social reality encompasses the agent's (i.e., potential individual migrant) face-to-face and indirect social interactions within or outside the household unit and with other groups and institutions (e.g., school, local administration, etc.) which are constrained by macro level forces (structures) to shape young peoples' migration aspirations and capabilities.
- Meso-level: This level of social reality focuses on institutions and social groups, namely household units (which include members of the extended family) which adopt remittance to be garnered as a household strategy for spreading risks or diversify livelihoods (in de Haas 2010). At the meso level, we argue, formal and informal institutions, social networks and friendship circles increase the aspirations and capabilities of agents (i.e., potential migrants).

- Macro-level: Refers to national policies affecting individual migrants and meso-level social units through increased "negative liberty" (de Haas 2014) (resulting from failing state policies, high unemployment of educated youth, generally poor macro-economic outlook).

We argue that young peoples' decisions to drop out of school in pursuit of their migration plans is an outcome of complex interactions between (a) individual choices that shape their aspirations and in turn attitude towards schooling and the perceived outcomes of migration; (b) support and approval by parents, extended family, and the larger community to the migrant; and (c) the discouraging and encouraging national socio-economic and political context. The embeddedness of migration aspiration and capabilities within the micro, meso, and macro-social spaces is underpinned by Turner's [38,39,51] macro-meso-micro sociology since at each level of social reality, agency and structure interact to allow or disallow individuals or households to make migration decisions, either by boosting the ability/capability of young people to migrate (i.e., positive liberty) or vice versa.

*Sampling and Data Analysis*

Whilst much migration research depends on a positivist epistemology, our study adopted a qualitative, interpretive approach to increase our understanding by describing the lived experiences of young people (mainly students) who plan to migrate (dropping out of school), as well as make sense of the subjectivities of their teachers, principals, parents and key members of the local communities. A limitation of this specific focus is that the findings are not generalizable to all individuals. Nonetheless, the findings still contribute to the ongoing conversation, providing detailed insight into one of the factors, which may feed into future studies that integrate a broader array of factors influencing choices to complete education, to consider international migration, or to explore opportunities domestically.

As indicated, the study sites were Hadiya and Kembata zones. These specific locations were chosen not only because of high migration rates, but also due to the low average school performance and high dropout relative to other areas within the Southern Nations, Nationalities and Peoples' regional state. The total number of respondents that participated in the interviews was 18. These included: eight students (four male, four female) who were aspiring to migrate, three school principals in the target location, five parents (three in Hadiya and two in Kembatta), and four teachers of which, one was a college lecturer (who formerly served as school principal in two different locations of the study area); the latter was included because he previously worked as a secondary school principal and had in-depth knowledge of the problem under question. Data were generated using semi-structured interview questions addressing the values' community members attach to education, the factors that affect their views about educating children, and their perspectives about how migration and education are affecting each other in the context of the study. The data gathering process was started after obtaining ethical clearance by the proposal review committee of the Institute of Policy and Development Research (IPDR) at Hawassa University.

Having completed the field work, the data generated from semi-structured interviews were transcribed and edited for accuracy of the Amharic-English translations. The transcripts were then categorized based on the re-current themes that emerged. Taking a qualitative approach, this study presents opportunities and limitations. The opportunities are the resulting wealth of insight about the questions posed, while the limitations relate to the generalizability of those insights. We offer the findings of this paper recognizing both the opportunities and limitations.

## 5. Results

Based on the conceptual framework described above, the results are presented under the following loosely structured categories: (1) individual assessment of education as a capability, (2) social construction of capability (and, in turn, functionings) as a determinant of migratory agency, and (3) structural constraints in migration decisions. An additional section analyzes the gendered dimensions of migration and the gendered patterns of migration decisions. Although not a focus of this study per se, it is worthwhile noting that the micro, meso and macro environments are dynamic. We have not

offered a longitudinal study of these changes, but rather reflect upon these changes as they relate to each level as they are raised by interviewees.

*5.1. Individual's Assessment of Education as a Capability*

Respondents felt that a significant change had occurred over the past few decades. In the past, according to one of our respondents, "children struggled to enter school and seek the benefits offered by it, but today this level of eagerness doesn't exist". One emerging alternative is the allure of international migration for opportunities that offer higher pay than what would be available for them after competing secondary school. A principal we interviewed recalled a student boldly stating:

When he [the teacher] asked him: 'Why aren't you interested in school?' Can you guess what the student said? You would never believe! He firmly replied to his teacher: 'Why should I worry about education if studying hard and finishing college is to end up like you? I am sure I will see you teaching in the same school as poor as you are now when I get back home with a lot of cash!' SP-A

Similarly, a teacher observed:

Children in our locality have grown up watching the lavish weeding videos and pictures sent from locals in the diaspora, especially from South Africa. I and the majority of the local people believe that this has captured the imaginations of many young people . . . everybody dreams to go out! . . . When they are back on vacation [migrant locals] they show-off their valuable possessions . . . they have all sorts of gadgets: smart phone, camera, nice clothes, etc. Young people are highly attracted by these things! They [youth] dream, that one day, they would be able to do the same.

However, the opinion of another teacher altered the causal relationship of these factors. Rather than see little value in education at the outset, those students who do not perform as well as others adopt migrating out as a viable alternative for them, based on their future prospects. Explaining that the "majority of young people who opt to migrate are less academically competent and see little prospect of joining higher education." Regardless of the cause, the values regarding opportunities have changed. A teacher noted:

Mostly female students migrate to Middle Eastern countries while males prefer to go to South Africa . . . For them, it is a big deal to go abroad no matter the consequence! Boys bring their passports to school to boast around their fellow classmates!

The decline in the value that people attach to education, especially among students, is manifested in various ways including showing low respect for their teachers. It is also manifested in their aspirations being only minimally linked to their work and success in school. On the other hand, one cannot conclude that households (parents) and young people dismiss education as unimportant because they still realize that achieving a certain level of education is critical, if not mandatory to be able to migrate. The aspiration–capability framework posits that people's encounter with successful migrant returnees and members of the diaspora could enable them to get some knowledge and information about the benefits of migration, which, in turn, increases their aspirations and capabilities to migrate [12].

*5.2. Social Construction of Capability*

There is a saying *'yetemare yet derese'*, meaning educated people have not succeeded. This is because there are graduates who did not find work—SP-A

Migrating out of the country has now become a culture in our community! People [parents and the community] support the idea of girls going to Middle East, and boys to South Africa with higher regard than succeeding in school—SP-C

Parents and community members engage in social comparisons that alter collective aspirations in a way that gives little value to academic engagement. From the findings, it is evident that boys and girls who have managed (but who did not have much education or no education at all) to migrate to South Africa or to the Middle East were able to demonstrate their success financially by accumulating wealth. In so doing, they are highly respected and seen as role models that are worth emulating. One of the key reasons for this, according to one of the school principals is:

> These days, going out of the country is far better than earning university degree ... I had some students who dropped out in grades 6 and 7 who are now in South Africa; they were able to build a better house and to buy cars. Civil servants are still living in rented [do not own] houses . . . . This is the reason why young people migrate, leaving their education aside.

> It is commonplace to hear people say: "what did those children do to their parents after staying all these years in school?" Look at what a certain person's daughter or son bought to her/his father? Didn't you see the big house his/her daughter built? Haven't you seen the Isuzu truck she bought? SP-B

On the other hand, we found evidence where education has contributed positively in terms of increasing the ability (capability) of the local youth to migrate. A teacher explains:

> Young people who have some high school education know how to use the Internet ... those who have little or no education are not even aware that they can communicate with their relatives outside the country. They frequently go to Durame [nearest town] where they can find Internet Cafés. Because of this, better educated youth often easily plan their journey.

However, though one could argue that increased education is related to increased capability, education alone does not result in migration. There are other factors influencing these choices.

One of the critical changes is rapid urbanization. Within the studied communities, urbanization manifests itself through growing peri-urbanization areas and the incorporation of rural land into it. Due to population growth, there is a declining size of farmland per capita. The realization that not all children will inherit sufficient land (or any at all) forces youth to reconsider their future options [52]. Urbanization adds another layer to this process, as families lose their land to the expansion of urban areas. This includes an array of processes—from infrastructure development and business expansion to land speculation [53]. As the urban areas become more accessible, youth are exposed to new options, opportunities and ideas, thereby altering their vision of what a 'good life' encompasses [12]. De Haas [11] argues that, while such capabilities and aspirations manifest themselves on an individual level, they are affected by macro-structural changes such as the development of infrastructure, schooling and media. From this, it seems that people are most inclined to migrate when they enjoy a high degree of negative freedoms and 'moderate' level of positive freedoms, which should be high enough to enable people to migrate, but not so high that declining spatial opportunity differentials with potential destinations would substantially decrease migration aspirations.

Our findings are not only in agreement with the theoretical perspective advanced by the New Economics of Labor Migration (NELM), which posits migration as a strategy for risk aversion, but more broadly with the aspiration–capability perspective. Specifically, the interviews with our respondents show that potential migrant households and individuals feel a sense of marginalization or exclusion from economic and political decisions due to failed policies accompanied by poor governance practices at local and national levels. The feeling of exclusion/marginalization by government policies (macro-level) and the manner in which they are implemented by local political and government institutions (meso-structure) exerts "negative liberty" (capability), while, at the same time, fueling the migratory agency of individuals (and household units). This illustrates what de Haas [12] (p. 29) refers to as the "dialectics of structure and agency in the process of migration."

Consistent with the New Economics of Labor Migration, the findings show that the desire to migrate (terminating school) is found to depend on a household's decision in the context of the

communities studied. In other words, regardless of the level of an individual's "migratory agency", decisions about his/her migration are made by the household unit. Migration decisions are not made solely by an individual member of a household, but through collective decision making of a household as a social unit because migration is a household strategy [16] to ensure the economic survival for all members of the household.

As can be evident from the forgoing discussion, one cannot conclude that parents and other members in the studied communities harbour negative attitudes toward education. Nor could one argue that those parents who uphold the values of education could spare their children from irregular migration. In fact, what underlies the positive attitude towards education among some of our respondents (which included parents, teachers, and students) seems to resonate with the reasons why parents in East Asian (i.e., India, Vietnam, and China) countries opt to educate their children, as Punch and Sugden [54] reported. Thus, like East Asian parents, the emphasis on education is associated with the parents' desire for their children to escape the difficulties of traditional rain-fed agriculture, not to mention the declining size of farmland in the target communities due to population pressure and environmental degradation as reported in most recent studies [27].

However, willingness to invest in children's education seems to decline overtime owing to the value that parents attach to education, especially in terms guaranteeing a decent life for young people in their localities and beyond because, as our findings show, parents justify their investment not only in terms of gainful employment for their children, but also as "a retirement plan" for themselves (i.e., children give back through financially support). For their parents, it is frustrating to witness the fact that even college educated youth could not find gainful employment. When they do, the earnings are too little to put food on the table for themselves, let alone to support their parents as the latter originally planned. In this vein, while some of our respondents are still of the opinion that wider job opportunities could help curtail mass migration, we found counter evidence. This finding suggests that even employed young members of the community (e.g., teachers and civil servants) have left their jobs to migrate to South Africa. One respondent explains the logic: The basic salary of civil servants and teachers per month for a degree graduate after tax deductions is 3250 Ethiopian Birr (ETB) (approx. 118 USD) which does not cover the basic costs of living (e.g., paying house rent, food and other basic expenses). Thus, it is unthinkable for a young professional or degree holding civil servant to plan to rent a single-bedroom service quarter of a modest tin-roofed villa, let alone to own one. The interviewee went on to stress that government employees are not gainfully employed (i.e., under-employed) either. As a result, despite being employed, they opt to migrate. Here, it is evident that education does not guarantee the life young people reason to value because, compared to the uneducated, the educated youth showed more dissatisfaction of wider job opportunities due to their qualifications or increased access to current information that an educated person could take advantage over the uneducated. Instead, the majority of our respondents believe that people with little or no education, yet own very small businesses (e.g., street vendors, taxi drivers etc. . . . ), in many cases earn better than college educated government workers. One interviewee reflected:

> Once my uncle said: 'Your dedicated service for more than 15 years as college/university teacher is by far less than your secondary school dropout nephew who stayed only for 3 years in South Africa'. He built a villa for his aging parents in Durame [Town]!' That was very painful to hear from your relative!

As a result, households not only engage in risk spreading, but also take additional risks through borrowing huge sums of money to finance the costs of migration. A student explains:

> When my brother illegally migrated to South Africa in 2015, he had to drop out of school. Our parents did not have enough money to pay for all expenses. So, they had to sell three oxen and other valuable things were not enough to cover the costs! Then, they [parents] had to borrow a lot of money with high interest. My brother is still paying this debt.

*5.3. Structural Constraints in Migration Decisions*

Our interviewees indicate that the positive attitude towards education did not translate into the actual earning that they hoped would result from their years of investment of educating their sons and daughters. Closer scrutiny of the interview findings show that the responses could be broadly classified into two, namely (1) the quality of education and training and (2) the relevance of educational programs (see [55,56]. These two structural (macro) factors are found to boost "migratory agency" in rather negative ways. While poor quality of education results in unemployable technical school and university graduates, the lack of relevance (of educational programs) makes the demand for the knowledge and skills acquired by graduates of limited value in the current labor market. A study in Addis Ababa similarly showed that young people who intend to migrate internationally were graduates of TVET colleges and universities whose qualifications had low demand in the job market [55].

We frame the education–migration nexus within the CA based on the assumption that education can effectively play its capability extending roles as long as minimum quality standards are met [18,33] and equal opportunities exist that take into account the diversity of social groups who would require varying degrees of access to resources to convert capabilities to real achievements (functionings). As such, equal access to minimally acceptable quality of education [33] and an education that is relevant to people in the target communities are viewed from the point of view of ensuring social justice [33,35,57]. We, therefore, argue that inability to achieve valuable functioning (e.g., gainful employment), despite completing a certain level of education, forces young people to make (risky) decisions of perusing irregular international migration. In other words, there is a systemic policy failure of the national system of education and training. The system has been graduating large numbers of new graduates every year (about 150,000) [56], most of whom are unemployable. Consistent with other research [35], the inability of graduates to achieve functioning (gainful employment or self-employment) could be partly attributed to social inequality. Young people in the target communities are mainly rural or peri-urban areas where access to quality education is limited. The inequality of opportunities in accessing (quality) education was mentioned by our respondents in relation to the inadequacy of resources and teaching, learning materials and facilities, as well as poorly qualified and demoralized teachers. According to available evidence [36], such structurally imposed constraints generate inequalities and reduce the capability of young students to achieve what they want to achieve. In contrast, education has helped to bolster what de Haas [11] refers to as "migratory agency"—not because the respondent received a good quality and "functionality" through boosting employable knowledge and skills, but only because it increases the level of awareness of the youth about the benefits of migration out of the country (mainly to the Middle East and South Africa).

The meso- and macro-level political structures are not monolithic. From an educational point of view, the trends described in this paper are negative. However, migrants that successfully reach new destinations often send remittances and savings to their home communities, starting business and making investments. This positive economic impact is one reason the Government of Ethiopia has sought to improve some forms of migration and capitalize on the positive economic impacts of those collective choices. The result could be viewed as a macro-level policy incoherence [58], whereby economic policies seek to encourage and facilitate migration while educational ones seek to ensure children and youth complete their education.

Community members see the positive impacts migration has upon individuals and households:

> Successful migrant returnees open small business such as hotels and restaurants, schools, etc. They are employing local people and their relatives. By sending remittance migrant youth are helping their relatives and siblings to go to good schools ... They are able to educate [pay for] their children or siblings in the family better; they support their parents and they build new houses for the family.

Alternatively, a father comments that: "people who do not have relatives, are facing a lot of problem including shortage of food. Some of them do not even have any cattle; they cannot change

their house [grass covered mud huts] either; their life is vulnerable." According to the household theory of migration, family units send members of the household to work in the global labor market in order to increase flow of income and to decrease economic risks of the family [15,20,59,60]. Indeed, research on the topic has repeatedly demonstrated that earnings of labor migrants are considerably higher than the earnings they had prior to migration [11,59,60] and that migrants remit considerable portions of their earnings back home [60]. Therefore, earning gains and the ability to remit back home are often viewed as the major motivations driving global migration [59,60]. This is consistent with Andersson's [43] study in an Ethiopian context, which reveals that households receiving remittance have higher subjective wellbeing than non-receiving households.

### 5.4. Gendered Education–Migration Nexus

The decision to drop out of school and migrate is not always in the best interest of the migrant. This is particularly the case when the migrant is a female. In this regard, the findings based on respondents' interviews seem to give empirical support to feminist critiques of migration that cultural norms in the sending and receiving countries play a critical role [61–63] in explaining the gender differential in international migration [62].

Our in-depth interviews with school principals and female students, particularly in Hadiya Zone corroborates the feminist view. The response of the Principal in Gimbichu outlines how female students are forced to drop out of school to migrate out of the country to fulfill the needs of parents and the extended family. For example, a grade 12 female student reported the undue pressure exerted on her close friend from own parents and relatives to go abroad. She explained:

> I remember that I had a classmate when I was in grade 8. Her parents forced her to migrate to one of the Arab countries because they [parents] wanted her to improve their life with the money she sends back home. By the time she dropped out of school and left, she was about to write the 8th Grade National Exam. But, since she was trafficked illegally, after a lot of suffering along the way she returned five months later without reaching her destination . . .
> By then, she was physically and morally destroyed. Though she resumed school, her lagging behind her friends, and the bad experience, which included rape, had affected her in a negative way.

A principal (SP-A) further explains that "parents worry very little about the psychological consequences . . . . Those [young women] who returned from Middle Eastern countries are suffering the trauma, the feeling of emptiness and broken self-esteem."

Even though migration decisions, by and large, were found to be made by households, in both target communities, gender differences were evident in terms of the geographical locations of the destination countries. According to the majority of respondents, females primarily migrate (or plan to migrate) to the Middle East (such as Bahrain, Egypt, Lebanon, Saudi Arabia, Qatar) while males mainly moved to South Africa. The reasons resonate with gender socialization theory, which posits that a division of labor between men and women in patriarchal societies is line with sex-appropriate traditional roles where women assume the task of housekeeping and taking care of children and the elderly. Accordingly, the majority of the respondents said that, as opposed to men who tend to assume more visible roles, women are in high demand for domestic work in Arab countries where they would be less visible and experience a high potential for abuse and exploitation [61]. Notwithstanding this, households encourage their daughters to migrate (legally or otherwise) and start remitting as soon as possible. On the other hand, males are actively encouraged (by parents, relatives and friends) to migrate to South Africa for at least two reasons: (1) the kind of employment requires traditionally masculine work, involving hard labor often outside the domestic sphere of adult or child care and serving family members; and (2) there are a significant number of migrant communities in South Africa (some of which running private business) who call for male laborers from their ethnic in-group. Given that such businesses involve risk of violence and attacks (as often the case against Ethiopian

communities in South Africa), they call for men to make the journey. If females migrate to South Africa, this tends to be either to join their relatives or kin, or following marriage.

## 6. Conclusions

Contemporary theories of migration emphasize the role of education (credentials) as "capability" enhancing and as an additional input in terms of increasing successful prospects for migrants. Despite this positive assessment, young people (with the endorsement of the household) opt to abandon (terminate) their education in order to migrate internationally. Our study attempted to unravel some of the underlying causes why this is happening in the studied communities and what factors underpinned migration decisions. The findings generally suggest that a complex set of factors come into play. Nonetheless, none of these factors neatly fall in any of Turner's [39,40] levels of social reality. This provides empirical evidence to his theoretical proposition that the levels of social reality are embedded within each other. Specifically, individuals were not the sole decision-makers for the youth to migrate abroad; nor are members of the extended family, or members of social networks alone in destination countries.

Since capabilities are socially constructed [64], the value of education is as good as how institutions and society views or weighs its currency. Thus, a person's 'functioning' is not directly proportional to the level of educational qualification achieved. Rather, it is worth as much as how society construes the value of the qualification based on what is meant to be functional for the society in question. The findings based on our interviews with the respondents suggest that, at least, the instrumental dimension of education, has not fully enabled the youth to achieve "functionings" [38]. We argue that the decision of young people to terminate (i.e., dropout) school in order to migrate is not a decision made by individual migrants, but it is a collective decision of the individual, parents, members of the extended family, and the community; and the macro-level social reality represented by the national socio-economic and political context.

Further, the results generally suggest that yearning to migrate not only outweigh the desire to complete the level of school that potential migrants have already started, but also the motivation to excel in school work. Counterintuitively, in the long-run, migration is helping to improve educational opportunities for those who are left behind. This is evident from the findings that some migrants are contributing to educational development from paying school fees to ensure their siblings and relatives continue their education, establishing private schools that spill over to communities. This appears to leave conceptual difficulties particularly in terms of interpreting the notions of 'migratory agency', aspiration, and negative liberty' or positive liberty' since the aspiration–capability framework seems to take account of the agent (individual) rather than the collective (a household). Hence, there is a need to readjust the existing conceptualization to account for some key elements of NELM. The present study reveals that the relationship between education and migration is rather complex and far from linear.

According to Bourdieu [65], education not only helps to accumulate cultural capital, but also is instrumental for upward social mobility. As a cultural capital, education is convertible to economic capital that creates social inequalities. However, our findings contest Bourdieu's analysis since we found that education failed to maintain its ability to create cultural capital, and in turn economic capital. As such, what communities and parents would have expected to get from college educated (yet unemployed) youth is effectively replaced by the income that flows from abroad by less educated migrant members of the household. Thus, in what appears to be consonant with social network theory, families or households in the studied communities have fully adopted international migration as a strategy of rational allocation of family resources and increase the flow of income through remittance [66–68]. The findings confirm this theoretical standpoint.

In addition, the results show that migrants from semi-urban and rural villages tend to be employed in secondary jobs that do not entail a high level of education. In most cases, those migrants were destined for South Africa (usually male), and tended to be employed in ethnic businesses or become street vendors often as part of a network of a common social or ethnic origin. Similarly, migrants

to Middle Eastern countries (mostly females) usually end up working as housemaids. For potential migrants, therefore, getting more education does not make much difference in terms wage differentials. This is because they would be employed in areas that do not require more than basic literacy and numeracy. For those who do not plan to migrate, getting higher education qualifications (like getting a degree) might help them to be employed elsewhere, but does not necessarily guarantee sufficient earning to make ends meet. However, the opportunities that follow from obtaining education have helped to increase migration aspirations via boosting the capabilities to migrate. For example, migrants to South Africa and Middle Eastern countries need the TVET Certificate (Grade 10 + 1 or Grade 10 + 2) to secure their driving license as evidence for possessing the required skills in the destination. It follows that potential migrants will prefer to wait until they complete a certain level of education needed to do so. In that sense, the empirical link between education and migration is far from linear. To adequately understand the education-migration nexus a range of contextual factors should be accounted for across cultures, socio-economic and geographical realities with help in-depth qualitative studies.

**Author Contributions:** T.S. is responsible for the conceptualization, methodology, and investigation; L.C. and T.S. together did the formal analysis and writing.

**Funding:** This research received no external funding.

**Conflicts of Interest:** The authors declare no conflict of interest.

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
