# Peer review of "Education—Migration Nexus: Understanding Youth Migration in Southern Ethiopia"

_education, doi:10.3390/educsci9020077_

Round 1

Reviewer 1 Report

Thank you for the opportunity to review this manuscript. I think it is an important and timely topic. The author(s) have taken an interesting approach to the topic which, as they correctly point out, is quite novel. Content-wise, the article is strong and I recommend it for publication.

However, there are some serious style issues in my opinion. I suggest the author(s):

- thoroughly copy-edit for readability (long, confusing sentences with sometimes contradictory points; circular arguments; etc)

- thoroughly copy-edit for minor type-os (incl. cited author names etc)

-thoroughly copy-edit for style (major points are hidden in paragraphs. for example, your 'driving research question' is in the middle of a paragraph - line 246. The readers shouldn't have to read it with a fine tooth comb to understand what exactly you're asking.

- reframe opening paragraphs to recognise this is an application of de Haas (2010) rather than repeat his logic and come to the same conclusions. You can recognise it as such, then explain why it's appropriate. You do a masterful job making the argument that you're building on de Haas, but then conclude as de Haas. Why not just say using de Haas, we look at the aspirations-capabilities framework to understand ... and then go into the reasoning and what you hope it will uncover?

- the method section needs more clarity. The focus is on what drives young people, and while it absolutely fits your framework to ask adults, make the rationale explicit and clear. Link the methodology to aspirations-capability and to the micro, meso, macro analyses. 

- double check your claims/conclusions etc can't be interpreted in other ways and make the case for the way you're interpreting them. For example, is it a decline in the value of education (as you state) or a decline in the value of opportunities in Southern Ethiopia for them? When making this interpretation, be mindful of the aspirations-capabilities framework you've set up.

While these are fairly extensive suggestions, I do highly recommend your work and look forward to seeing it published. I believe these revisions will greatly strengthen the article.

Author Response

Response to Comments and Suggestions to Reviewer I

First of all, we would like to thank you for your insightful comments and suggestions on our manuscript. We benefited a lot from your constructive and educative comments. We tried to address your concerns as much as possible. Below are our responses and revisions on the original manuscript as stated by Reviewer I. 

1.      - thoroughly copy-edit for readability (long, confusing sentences with sometimes contradictory points; circular arguments; etc)

We did substantial copy-editing to improve the readability of the manuscript. We went through the manuscript between the lines to weed-out sentences or phrases that are unclear or confusing.

2.      thoroughly copy-edit for minor type-os (incl. cited author names etc)

As we did the language editing we also addressed minor typos. Further, we checked the accuracy of authors’ names and dates in in-text citations, and accuracy of bibliographic information in the list of references.

3.      -thoroughly copy-edit for style (major points are hidden in paragraphs. for example, your 'driving research question' is in the middle of a paragraph - line 246. The readers shouldn't have to read it with a fine tooth comb to understand what exactly you're asking.

The research questions are restated such that the to improve ease of reading.  Similarly the paragraphs in the introductory section are re-written.

4.      reframe opening paragraphs to recognise this is an application of de Haas (2010) rather than repeat his logic and come to the same conclusions. You can recognise it as such, then explain why it's appropriate. You do a masterful job making the argument that you're building on de Haas, but then conclude as de Haas. Why not just say using de Haas, we look at the aspirations-capabilities framework to understand ... and then go into the reasoning and what you hope it will uncover?

As per the above suggestion, we started with a new paragraph (see: line 48-53) 

The present study employs de Haas’s (2010) aspiration-capability framework to understand  the education-migration nexus in southern Ethiopia. Specifically, we examine the reasons behind young peoples’ decision to give up school at micro, meso, and macro levels (Turner 2010a; 2012). The findings inform decision makers regarding (re)designing strategies to reduce the negative ramifications of irregular migration. The depth of qualitative data also informs future research, particularly what measures ought to be considered when conducting quantitative studies.

5)     the method section needs more clarity. The focus is on what drives young people, and while it absolutely fits your framework to ask adults, make the rationale explicit and clear. Link the methodology to aspirations-capability and to the micro, meso, macro analyses. 

As shown below, we re-worked the methodology section by taking account of the above suggestions. In addition, the locations of the paragraphs are changed to improve readability  (see Section 4)

Materials and Methods

Within Ethiopia, there are regions where formal and irregular migration is more common than others. For example, migration to the Gulf Cooperation Countries is a common practice of youth in Tigray regional state while in parts of the Southern Nations, Nationalities and Peoples’ (SNNP) regional state there are higher rates of migration to South Africa (mostly through Kenya, Tanzania, Malawi, Mozambique) and the Middle East (Berhe 2013; ILO 2017). There are other locations from which migrants originate (Amhara and Oromia regional states), but limited data is available about the destinations (ILO 2017). This study focuses upon two locations within SNNP regional state, namely Kembata-Tembaro and Hadiya zones. Specifically, this study analyzes the linkages between education and migration. The research questions we set out to answer were: (a) What is the role of education in influencing decision making for youth to consider migrating out of the country? Does education contributes for increasing aspirations and capability in migration decision?

We applied de Haas’(2010) aspiration-capability framework to evaluate if education is enabling youth to achieve the functionings required: (a) to take advantage of opportunities available to them domestically? and (b) the extent to which education, as a core capability (Nussbaum, 2003),  impacted (positively or negatively) on young peoples’ migration aspirations. We closely examine these questions by specifically focusing on young people who aspire to migrate out of the country terminating their education. Since migration is a sociological phenomenon, we conceptualized the interaction between aspiration and capability in shaping migration plans (or actual decision to migrate for that matter), using Turner’s (2010a; 2010b; 2012) integrated sociological theory as our analytical lens, we account for the micro-meso-macro levels of social reality. Thus, the selection of different categories of respondents i.e. students, teachers, school principals; parents, and other knowledgeable community members was meant to understand how decision to migrate by an agent (individual migrant) is viewed and judged at micro, meso, and macro levels of social reality.

More specifically, the micro-meso-macro enables us to explore who makes the decision in order that the young person giving up school to migrate; what roles the individual migrant, his/her parents and members of the extended family, neighbors, the village community, the local governance practices play; and the national socio-economic and political (macro) context plays in feeding into the migration aspiration and migration capabilities in making the decisions. Drawing on Turner’s (2010a; 2010b; 2012) macro-meso-micro theories of sociology, we operationalized the levels of social realities as follows:

·        Micro-level – the micro level social reality encompasses the agent’s (i.e. potential individual migrant) face-to-face and indirect social interactions within or outside the household unit and with other groups and institutions (e.g., school, local administration etc.) which are constrained by macro level forces (structures) to shape young peoples’ migration aspirations and capabilities.

·        Meso-level: This level of social reality focuses on institutions and social groups, namely household units (which include members of the extended family) which adopt remittance to be garnered as a household strategy for spreading risks or diversify livelihoods (in de Haas 2010). At the meso level, we argue, formal and informal institutions, social networks and friendship circles increase the aspirations and capabilities of agents (i.e. potential migrants).

·        Macro-level: in this paper refers to national policies affecting individual migrants and meso-level social units through increased “negative liberty” (de Haas 2014) (resulting from failing state policies, high unemployment of educated youth, generally poor macro-economic outlook).

Thus, we argue, that young peoples’ decision to drop out of school in pursuit of their migration plans as an outcome of complex interaction of (a) individual’s-choice that shapes their aspirations and in turn attitude towards schooling and the perceived outcomes of migration; (b) support and approval by parents, extended family, and the larger community to the migrant; and (c) the discouraging and encouraging national socio-economic and political context. Accordingly, other things being constant, individual students might decide to drop out of school in pursuit of migrating outside the country. The embeddedness of migration aspiration and capabilities within the micro, meso, and macro-social spaces is underpinned by Turner’s (2010; 2012) macro-meso-micro sociology since at each level of social reality, agency and structure interact to allow or disallow individuals or households to make migration decision either by boosting the ability/capability of young people to migrate (i.e. positive liberty)

4.1.Sampling and data analysis

Whilst much of migration research depend on positivist epistemology, our study adopted  qualitative, interpretive approach to increase our understanding by describing the lived experiences of young people (mainly students) who plan to migrate (dropping out of school); as well as make sense of the subjectivities of their teachers, principals, parents and key members of the local communities. A limitation of this specific focus is that the findings are not generalizable to all individuals. Nonetheless, the findings still contribute to a conversation, providing detailed insight into one of the factors, which may feed into future studies that integrate a broader array of factors influencing choices to complete education, to consider international migration, or to explore opportunities domestically.

As indicated, the study sites were Hadiya and Kembata Zones. These specific locations were chosen not only because of high migration rates, but also due to the low average school performance and high dropout relative to other areas within the southern nations, nationalities and peoples’ region. The total number of respondents participated in the interview were 18. These included: eight students (4 male, 4 female) who were aspiring to migrate, three school principals in the target location, five parents (three in Hadiya and two in Kembatta), and four teachers of which, one was a college lecturer (who was formerly served as school principal in two different locations of the study); the latter was included because he previously worked as a secondary school principal and believed to have in-depth knowledge of the problem under question. Data was generated using semi-structured interview questions addressing the values community members attach to education, the factors that affect their views about educating children, and their perspectives about how migration and education are affecting each other in the context of the study. The data gathering process was nevertheless started after obtaining ethical clearance by the proposal review committee of the Institute of Policy and Development Research (IPDR), Hawassa University.

Having completed the field work, the audio data generated based on semi-structured interviews were transcribed and edited for accuracy of the Amharic-English translations. The transcripts were then categorized based on the re-current themes emerged. Taking a qualitative approach, this study presents opportunities and limitations. The opportunities are the resulting wealth of insight about the questions posed, while the limitations relate to the generalizability of those insights. We offer the findings of this paper recognizing both the opportunities and limitations.

- double check your claims/conclusions etc can't be interpreted in other ways and make the case for the way you're interpreting them. For example, is it a decline in the value of education (as you state) or a decline in the value of opportunities in Southern Ethiopia for them? When making this interpretation, be mindful of the aspirations-capabilities framework you've set up.

The conclusion section is partly re-written to make sound interpretations. Besides, the aspiration-capabilities framework is taken into account along with other relevant models. Based on our findings, we also made suggestions to make aspiration-capabilities framework more inclusive.   

While these are fairly extensive suggestions, I do highly recommend your work and look forward to seeing it published. I believe these revisions will greatly strengthen the article.

Definitely, we have benefited significantly! We are highly grateful for the invaluable comments and suggestion because, now, we feel the quality of the article has substantially improved

Reviewer 2 Report

This study strives to explore the education-migration nexus in Ethiopia by examining why young people terminated their education in order to migrate out of their country. It is a meaningful study reflecting economic development and educational issue in current African context. This paper was well-structured and well-written.

Abstract:

It stated that “It examines why young people migrate out of their country, terminating their education.” I think it should be change to: It examines why young people terminated their education in order to migrate out of their country. The purpose of this paper was to explore the reason why young people terminate their education rather than migrate out of the country.

In the abstract, sample size should be included.

Introduction:

There were lots of theoretical discussion at the section of Context: Education – Migration Nexus. They were rich in contents but might need to be revised to make them succinct. Some information might not necessary.

The same problem existed in section 2.1. Good information but might need to be better summarized.

Methods:

Turner’s theory of sociology was used in data analysis. I would suggest a brief summary of this theory before presentation of concept operationalization (see page 8).

Results:

Findings were well-presented and discussed.

Conclusion:

The conclusion was reasonable and meaningful.

Author Response

Response to Reviewer II

We thank you very much for critically reading our article and very helpful comments.

Abstract:

It stated that “It examines why young people migrate out of their country, terminating their education.” I think it should be change to: It examines why young people terminated their education in order to migrate out of their country. The purpose of this paper was to explore the reason why young people terminate their education rather than migrate out of the country.

We made the revision in the abstract, as suggested in the above paragraph

In the abstract, sample size should be included.

We specified the sample size in the revised version.

Introduction:

There were lots of theoretical discussion at the section of Context: Education – Migration Nexus. They were rich in contents but might need to be revised to make them succinct. Some information might not necessary.

·        We slightly reduced the introduction by removing sentences that are less relevant.

The same problem existed in section 2.1. Good information but might need to be better summarized.

·        As per the suggestion we reduced section 2.1 by 211 words.

Methods:

Turner’s theory of sociology was used in data analysis. I would suggest a brief summary of this theory before presentation of concept operationalization (see page 8).

For the sake of space and readability, we preferred not to give a separate account. Instead, we embedded the key concepts and constructs relevant to the study – and in different locations as deemed necessary. (Turner, 2010a; 2010b; 2012). Besides, due to extensive revision in the methodology section, we tried to address your concerns by providing more clarification on  the three levels of social reality and what makes Turner’s principles of sociology more relevant to understand social phenomenon like migration.